# Distribution and Health Risk Assessment of Trace Metals in Soils in the Golden Triangle of Southern Fujian Province, China

**DOI:** 10.3390/ijerph16010097

**Published:** 2018-12-31

**Authors:** Sha Huang, Guofan Shao, Luyan Wang, Lin Wang, Lina Tang

**Affiliations:** 1Key Laboratory of Urban Environment and Health, Institute of Urban Environment, Chinese Academy of Sciences, Xiamen 361021, China; shuang@iue.ac.cn (S.H.); shao@purdue.edu (G.S.); lywang@iue.ac.cn (L.W.); wanglin@iue.ac.cn (L.W.); 2University of Chinese Academy of Sciences, Beijing 100049, China; 3Department of Forestry and Natural Resources, Purdue University, West Lafayette, IN 47907, USA

**Keywords:** trace metals, the Golden Triangle of Southern Fujian Province, urban soil, ecological risk, health risk

## Abstract

In recent years, intensified industrialization and rapid urbanization have accelerated the accumulation of trace metals in topsoils of the Golden Triangle of Southern Fujian Province in China. Trace metals can cause adverse impacts on ecosystems and human health. In order to assess the ecological and human health risks of trace metals in the Golden Triangle region and to determine the distribution and degree of pollution of trace metals, 456 soil samples were collected from 28 districts. The concentrations of six metals (As, Cr, Cu, Ni, Pb, and Zn) were analyzed to assess ecological risk using the geoaccumulation index (*I_geo_*) and the potential ecological risk index (RI). The United States Environmental Protection Agency (USEPA) model was applied to calculate health risk. The average soil concentrations of the six elements are ranked as follows: As < Ni < Cu < Cr < Pb < Zn. Inverse distance weighting (IDW) interpolation maps showed that Cr, Cu, Ni, and Zn are enriched in the soils of developed areas, while As and Pb are enriched in the soils of undeveloped areas. The *I_geo_* showed that the levels of metals in most soil samples are below polluting levels. Similarly, RI values indicated that trace metals pose low potential ecological risk in the region’s soils. The Hazard Quotient (HQ) ranked the mean total noncarcinogenic risk of the six metals, for both children and adults, as follows: As > Pb > Cr > Ni > Cu >Zn. The mean carcinogenic risk (CR) of the metals in the region’s soils are ranked as follows: Cr > As > Ni. The Hazard Index (HI) values indicated that 3.7% of soils contained unsafe levels of toxic metals for children and total carcinogenic risk (TCR) values indicated that 23.3% of soils contained unsafe levels, indicating that children are facing both noncarcinogenic and carcinogenic risks from trace metals. Principal component analysis (PCA) and matrix cluster analysis were used to identify pollution sources and classified trace metals and soil samples into two and five groups, respectively. The five groups represented the effects of different land use types, including agricultural area, residential and public area, industrial area, forest, and industrial area and roadside, based on the contents of trace metals in soils. Industrial, agricultural and traffic activities attribute to the enrichment of Cr, Cu, Ni, Pb, and Zn in the region’s soils. Moreover, the accumulation of As and Pb are also attributed to atmospheric deposition. These results can contribute to a better understanding of pollution, ecological risks, and human health risks from trace metals on large regional scales like the Golden Triangle of Southern Fujian Province.

## 1. Introduction

With the acceleration of urbanization and industrialization, toxic contaminants are continuously and increasingly entering urban soils and causing environmental problems [1,2,3,4]. Urban soils are generally regarded as an important component of the urban ecological system, and excessive inputs of pollutants may deteriorate the soil ecological environment and change the physical and chemical properties of soils [2]. A more serious problem is that these pollutants may travel from urban soils to humans via various pathways (e.g., direct inhalation, ingestion, and skin contact absorption) and negatively impact human health [5,6]. A variety of toxic contaminants have accumulated in urban soils; trace metals and persistent organic pollutants (POPs) are the most concerning ones [7,8]. In recent years, trace metal contamination in soils has become a serious threat not only to plants and soil microbes but also to adults and children [7,8]. The sources of heavy metals are divided into two categories: natural soils are strongly influenced by pedogenesis, while urban soils are more profoundly disturbed by anthropogenic activities [9]. In reviews of heavy metal pollution in soils, anthropogenic sources have been found to include roadway and transportation emissions, waste water and gas from industrial activities, atmospheric deposition, sewage irrigation, pesticides, and fertilizers [1,10,11]. In recent years, human inputs have exceeded natural inputs as the primary cause of increases in trace metals accumulation in urban soils, especially in developed regions with high population densities and industrial activities [12,13,14].

Several published studies have focused on assessing the ecological and human health risks of trace metals by analyzing their concentrations in urban soils to evaluate pollution levels [15,16]. The most common methods used to calculate ecological risks are the geoaccumulation index (*I_geo_*), the Enrichment Factor (*EF*), the Potential Ecological Risk Index (RI), and the Nemerow Pollution Index (*PI_Nemerow_*). These methods each have their own advantages and shortcomings, so many studies have applied a combination of two or more methods for a more accurate and objective assessment. For example, several pollution indexes were used to determine the ecological risk of metals in the Planty Park soils in Poland [17]. Among these indices, *I_geo_* and RI were the most commonly used ones to assess the ecological risk of trace metals in the soils [18,19]. These two indices could assess not only metal soil contamination but also the degree of environmental risk [17]. Trace metals in the soils of urban and suburban areas can accumulate in the human body and threaten human health through three main exposure pathways: direct inhalation, ingestion, and skin contact absorption [7,11]. In most studies, the methods used to assess the harmful effects of trace metals depend on several factors, including the ages of affected individuals and the types of exposure to contaminated soils. As a metalloid element, arsenic has been included in many trace metals studies. One recent review assessed the pollution levels of eight common trace metals (As, Cr, Cu, Ni, Pb, Zn, Cd, and Hg) in the urban soils of 32 cities in China from 2006 to 2016 and evaluated their related human health risks [20]. This review found that, in most cities, trace metals in soils generally posed low noncarcinogenic and carcinogenic risks to the public, while in some industrial cities, the carcinogenic risk to children was a concern. Although urbanization and industrialization not only improve living standards, these processes accelerate the accumulation of pollutants, particularly the eight trace metals above, in urban soils, potentially elevating risks to human health [21,22,23]. For example, long term exposure of humans to As and Pb in soil may cause chronic disease [24,25]. Thus, these eight elements are commonly investigated in soil metal investigations. However, most research has been based on a single city or a small area within a city (e.g., urban parks or industrial area) [2,12,26]. With the acceleration of urban expansion, it is essential to focus new research on larger scales. 

At present, principal component analysis (PCA) is the most widely used method to determine latent factors indicated trace metals contamination in many studies [3,4,13]. In a review of studies using multivariate statistical analysis methods to evaluate soil heavy metal contamination, 66% of the studies used PCA to distinguish between potential natural and anthropogenic soil trace metal sources [10]. Moreover, in most studies, cluster analysis was the most common auxiliary method used to assist identification of the pollution sources [3,13]. However, matrix cluster analysis has rarely been used. Qing et al. applied matrix cluster analysis to divide the soil samples into four groups and distinguish the source of trace metals pollution in the urban soils of Anshan City [27]. This method not only classifies trace metals, but can also group sampling sites, which we think provides a more comprehensive approach to source identification.

The Golden Triangle is a metropolitan region with advanced economic and social development located in Southern Fujian Province, China. The most developed city in this region is Xiamen City, followed by Quanzhou and Zhangzhou City. In the coming decades, urbanization is expected to speed the rate of entry of pollutants from human activities into the urban ecosystem and cause an increasing number of environmental problems. However, few studies have assessed the ecological and human health risks from trace metals in Golden Triangle soils because of the challenges of sampling such a large land area.

Therefore, the main purposes of our study were to test the following three hypotheses: (1) trace metals accumulate in the soils of developed areas; (2) the ecological and human health risks from trace metals intensify in the soils of developed districts; (3) anthropogenic activities are the main sources of trace metals, which are identifiable by using PCA and matrix cluster analysis. Because the contents of Hg and Cd in many soil samples were too low to measure, we only analyzed As, Cr, Cu, Ni, Pb, and Zn in our study. To assess the ecological risk of trace metals in the soils of the Golden Triangle, two common methods, *I_geo_* and RI [17,18], were calculated in this research. The health risk assessment included two main aspects (noncarcinogenic risk and carcinogenic risk) and was based on three different exposure pathways (oral ingestion, inhalation, and skin contact) [2,18]. A comprehensive study focused on soil ecological and human health risks is urgently needed for the sustainable development large regions such as the Golden Triangle region. Our research findings can contribute much-needed reference values for controlling soil pollution and managing the risks of trace metals pollution in the Golden Triangle of Southern Fujian Province.

## 2. Materials and Methods

### 2.1. Study Area, Sampling, and Chemical Analysis

The Golden Triangle of Southern Fujian Province (Golden Triangle) is located in southern Fujian Province (longitude 116.530°–119.050° East, latitude 23.320°–25.560° North) and includes three cities: Xiamen, Quanzhou, and Zhangzhou (Figure 1) [28]. With an area of 25,315.39 square kilometers, the Golden Triangle accounts for one-fifth of the total area of Fujian Province. The study area is a mountainous and undulating coastal region, with the terrain of northwest at a higher elevation than the southeast (Figure 1). The climate is mainly subtropical with monsoon humid. Average annual temperatures range from 18 °C to 25 °C and average total annual precipitation is 1500–2100 mm. The Golden Triangle contains 40% of the population and 55% of the total economic output of Fujian Province, making it one of the most important cultural and economic regions in China.

In our research, we collected samples from all districts and the sample represented different types of land use to insure our results reasonable. We divided the total area of Zhangzhou and Quanzhou into 8 × 8 km sized grids on Google Earth, and one sample site was selected from each grid based on the land use and topographic conditions. Xiamen is the most developed city, but it is smaller than the others. The grid of Xiamen was 4 × 4 km. However, the Golden Triangle is mountainous region, especially Quanzhou City. Thus, the sampling sites were unevenly distributed. A total of 456 topsoil sites (0–20 cm) in 28 districts of the Golden Triangle were sampled between September 2016 and June 2017 using a Global Positioning System (GPS) to record the coordinates of each location (Figure 1 and Appendix A). These districts include Siming, Huli, Haicang, Jimei, Tong’an, and Xiang’an belong to Xiamen City; Fengze, Licheng, Dehua, Yongchun, Anxi, Nan’an, Luojiang, Jinjiang, Quangang, Hui’an, and Shishi belong to Quanzhou City; Xiangcheng, Longwen, Longhai, Hua’an, Nanjing, Changtai, Pinghe, Zhangpu, Yunxiao, Zhao’an, and Changtai belong to Zhangzhou City. At each sample site, four subsamples were collected from an approximately 16 m^2^ grid and thoroughly mixed to achieve a total composite soil sample weight of about one-kilogram. The soil samples were sealed in plastic bags and air-dried at room temperature (20–25 °C) for further analysis. After filtering through a 2 mm and a 2 μm polyethylene sieve in sequence, plant roots and shredded rocks were removed to prepare for chemical analysis.

First, each 0.20 g powered soil sample was digested with a mixed solution of HNO_3_-HF-HCLO_4_ (5:1:1) heated to 120 °C for 12 h on a heating plate. Then the residue remained was extracted with 0.5 mol/L HNO_3_ and prepared for quantification of As, Cr, Cu, Ni, Pb, and Zn [29]. To determine the concentrations of these elements, PEAA800 flame atomic absorption spectrophotometer (Perkin Elmer, Fremont, CA, USA) was used [30]. A separate bulk sample was tested after for every 10 analyses as an analytical control. A blank measurement was performed as the background value every 10 analyses. The analyzed precision was between 5% and 8%. Quality assurance and quality control (QA/QC) were established by determining the studied parameters in certified standard soil samples (soil GBW07405). The recoveries of six trace metals were ranged between 86% and 100%.

### 2.2. Statistical and Geostatistical Analysis

Before performing further analysis, the normality of trace metal contents should be checked. Log transformation and Johnson transformation were applied to improve the normality of these data (*p* > 0.05) using SPSS and Minitab 17. The statistical analyses included one-way analysis of variance (ANOVA), correlation analysis, PCA, and matrix cluster analysis. These were conducted with SPSS 19.0 for ANOVA and correlation analysis and R 3.2.1 software for PCA analysis. Correlation analysis was most often used to establish the relationships between the contents of the trace metals in the Golden Triangle topsoils; significance was set by the value of *p*, with *p* < 0.05 indicating a significant correlation and *p* < 0.01 indicating a highly significant correlation. In previous studies, ANOVA (*p* < 0.05) was typically used to determine the multiple correlations between different heavy metals contents. To categorize both the heavy metals and the 456 topsoil samples into different groups, matrix cluster analysis was the preferred method and was performed using statistical packages in R3.2.1. Geochemical maps of the trace metals were created using ArcGIS 10.1 (Redlands, CA, USA). Due to the complex terrain in Quanzhou, the sampling sites were unevenly distributed. Therefore, inverse distance weighting (IDW) was chosen to develop the geochemical distribution maps and interpolate soil trace metals concentrations between the irregularly-distributed sites [18].

### 2.3. Ecological Risk Assessment

#### 2.3.1. Geoaccumulation Index (*I_geo_*)

The *I_geo_* was used to determine the influence of anthropogenic factors on the levels of each trace metal in the topsoil samples [17]. The calculation was based on associated background values to calibrate the assessments. The *I_geo_* was calculated as Equation (1):
(1)$$I_{geo}=\log_{2}\left(C_{n}/1.5\times B_{n}\right)$$

The *I_geo_* results were divided into five classes, the corresponding pollution levels of the *I_geo_* classes is shown in Appendix A [31].

#### 2.3.2. Potential Ecological Risk Index (RI)

RI is another common approach to assessing the ecological risk of metal pollution and was employed in this study. The quantitative model includes the toxic response factor of trace metals (Tr) and the geochemical background value, as originally defined by Hakanson [32]. RI evaluates the potential ecological risk of all elements in the same sample, and was calculated as Equations (2) and (3):
(2)$$E_{r}=T_{r}\times C_{i}/C_{0}$$
(3)$$RI=\overset{n}{\underset{i=1}{\sum }}E_{r}$$

All of the abbreviations shown in these ecological risk assessment equations are described in Appendix A. Based on the RI values, the combined ecological risk of all trace metals was divided into four classes (Table 1) [1,18].

### 2.4. Human Health Risk Assessment

Human health risk assessment is a quantitative method for quantifying the adverse effects of human exposure to trace metals from a contaminated environmental medium (e.g., soils and sediments). Based on the Exposure Factors Handbook by USEPA [33], there are three main pathways for adults and children to become exposed to soil trace metals: oral ingestion, inhalation, and dermal contact. The health risk assessment method has been used in many studies focusing on risks to human health from trace metals in Chinese urban soils [20,34]. In the current study, the average daily intake (ADI) of soil metals via the three exposure pathways was calculated as Equations (4)–(6):(4)$$ADI_{ing}=\frac{C\times IR_{ing}\times CF\times EF\times ED}{BW\times AT}$$
(5)$$ADI_{inh}=\frac{C\times IR_{inh}\times EF\times ED}{PEF\times BW\times AT}$$
(6)$$ADI_{derm}=\frac{C\times SA\times CF\times SL\times ABS\times EF\times ED}{BW\times AT}$$

The definition, units, and reference values for all parameters used in the above equations are listed in Appendix A.

#### 2.4.1. Noncarcinogenic Risk Assessment

The Hazard Quotient (HQ) is a method designed to assess the noncarcinogenic effects of a specific trace metal in a given soil sample. It is the ratio of the average daily dose that a person receives through the three different pathways to a reference value RfD [20,27]. RfD is defined in Appendix A and is a reference dose for a given trace metal. If the level of a given metal is higher than RfD, then it may have harmful effects on human health. The Hazard Index (HI) is the sum of the HQ values for different metals and is applied to calculate the noncarcinogenic risk posed by multiple heavy metals. The equations are as Equations (7) and (8):
(7)$$HQ=\frac{ADI_{ing}}{RfD_{ing}}+\frac{ADI_{inh}}{RfD_{inh}}+\frac{ADI_{derm}}{RfD_{derm}}$$
(8)$$HI=\overset{n}{\underset{i=1}{\sum }}HQ_{i}$$

#### 2.4.2. Carcinogenic Risk Assessment

Carcinogenic risk (CR) is an individual’s lifetime risk of developing cancer from a potential carcinogen through the various exposure pathways. CR is defined as the risk from a specific metal and was calculated using ADI and the carcinogenic slope factor (SF) [20,27]. Similar to HI, total carcinogenic risk (TCR) is the sum of the CR values of different metals and was used to calculate the CR caused by multiple metals (Equations (9) and (10)).
(9)CR=ADI×SF
(10)$$TCR=\overset{n}{\underset{i=1}{\sum }}CR_{i}$$

The definition, unit, and reference values for all parameters used in the above equations are listed in Appendix A. HI and HQ values greater than 1 indicate a risk of suffering adverse health effects. CR and TCR values exceeding 10^−6^ and 10^−5^, respectively, indicate a risk of developing cancer [35,36,37].

## 3. Results and Discussion

### 3.1. Trace Metal Concentration in Golden Triangle Soils

Table 1 presents the descriptive statistics of six trace metals in the Golden Triangle soils and the corresponding background values in Fujian Province [38]. Figure 2 depicts the average contents of the six metals in Xiamen City, Quanzhou City, and Zhangzhou City. In addition, Figure 3 shows the spatial distribution of the six trace metals in the topsoils of the Golden Triangle of Southern Fujian Province. The mean contents of the six trace metals (As, Cr, Cu, Ni, Pb, and Zn) were 7.97, 22.80, 19.79, 10.51, 41.14, and 82.92 mg/kg, respectively.

To estimate the degree of influence from human activities on soil metal concentrations, the coefficient of variation (CV) of a given metal in the soils of the Golden Triangle has been used in many studies [27]. A high CV (CV > 50%) suggests that there are larger variations in metal concentrations among the different sites and that spatial distribution of the metal concentrations in the study area are non-homogeneous. The CVs of the six metals in the soils of the entire Golden Triangle region are as follows: Zn (53.73%) < Pb (63.61%) < Cr (85.09) < As (89.71%) < Cu (101.26%) < Ni (139.77%). The relatively high CV values indicate the non-homogeneous distribution of these trace metals concentrations [26].

In addition, the average concentrations of As, Pb, and Zn in the Golden Triangle soils were higher than the background reference values in Fujian Province, while Cr, Cu, and Ni were lower than the background values. The average values of trace metals in Quanzhou, Xiamen, and Zhangzhou were as follows: 7.82, 5.69, 9.72 mg/kg for As, respectively; 17.29, 33.58, 18.49 mg/kg for Cr; 23.29, 18.68, 18.34 mg/kg for Cu; 7.95, 10.41, 12.23 mg/kg for Ni; 37.73, 36.17, 49.96 mg/kg for Pb; and 98.22, 71.85, 81.10 mg/kg for Zn. In addition, the highest concentrations of As (96.6 mg/kg), Cr (148.1 mg/kg), Ni (153.8 mg/kg), and Pb (195.4 mg/kg) were found in Zhangzhou, whereas the highest concentrations of Cu (194.0 mg/kg) and Zn (296.0 mg/kg) were found in Xiamen and Quanzhou, respectively. These samples were mainly collected from industrial areas, representing industrial activity-affected surrounding soils.

In general, metal concentrations varied significantly across districts (Appendix A). Specifically, As concentrations in the Hua’an, Longhai, Dehua, Luojiang, Nanjing, Anxi, Yunxiao, and Pinghe district soils significantly exceeded those in Xiang’an, Tong’an, Siming, and Zhangpu (*p* < 0.05). By comparison, the districts with the highest Cr and Ni concentrations were Haicang, Jimei and Huli, indicating they might have similar pollution source (*p* < 0.05). The Cu levels in Yongchun district soils were higher than in the Tong’an, Luojiang, Xiang’an, Dehua, Zhao’an, and Dongshan district soils (*p* < 0.05). Yuxiao district soils had the highest Pb concentrations (*p* < 0.05). The Zn concentrations in the Licheng district were higher than in the Tong’an, Luojiang, Zhao’an, Yunxiao, Siming, Xiang’an, and Dongshan districts (*p* < 0.05).

GIS interpolation revealed continuous spatial distribution of metals in the Golden Triangle soils (Figure 3). Most areas of Zhangzhou City had high As concentrations in soils, especially in the Nanjing, Pinghe, Yunxiao Zhao’an, and Longhai districts. The concentrations of As in Xiamen soils were not as high as in the Zhangzhou soils. In contrast to the As levels, Cr levels were high in the majority of Xiamen soils, which lies at the center of the Golden Triangle, and in the Jinjiang and Longhai districts. The geochemical map of Ni showed that the high concentration of Ni was found along the coast, which was partial similar to Cr. Cu and Zn had similar spatial patterns, with higher concentrations in Quanzhou and Zhangzhou and lower concentrations in Xiamen. The distribution of Pb was different from Cr, Cu, Ni, and Zn, with low concentrations in the middle and northeast of the Golden Triangle, indicating that Pb soil pollution was more severe in Zhangzhou than in Xiamen.

### 3.2. Trace Metals Ecological Risk Assessment in Golden Triangle Soils

To assess the ecological risk from trace metals in the soils of different Golden Triangle districts, *I_geo_* and RI were evaluated (Appendix A). The range of *I_geo_* values for the six metals were: −2.53 to 3.48 for As, −5.95 to 1.26 for Cr, −4.34 to 2.58 for Cr, −2.54 to 1.90 for Pb, −4.50 to 2.93 for Ni, and −3.24 to 1.25 for Zn. The average values of *I_geo_* were less than 0 and decreased in the following order: As (−0.36) > Pb (−0.57) > Zn (−0.76) > Cu (−1.21) > Ni (−1.38) > Cr (−1.89). Most samples showed no signs of contamination (73.25% for As, 96.05% for Cr, 84.21% for Cu, 77.85% for Pb, 93.64% for Ni, and 85.96% for Zn), while 23.03%, 3.29%, 12.72%, 18.86%, 4.82%, and 12.72% of soils were slightly polluted with As, Cr, Cu, Pb, Ni, and Zn, respectively. Relatively few samples were moderately contaminated (2.41% for As, 0.66% for Cr, 2.63% for Cu, 3.29% for Pb, and 1.32% for Zn). Only 1.10% of samples were moderately to heavily polluted with As, 0.44% with Cu, and 1.54% with Ni, and only 0.22% were highly polluted by As. As shown in Appendix A, the mean *I_geo_* values of As in Anxi, Dehua, Hua’an, Longhai, Luojiang, Nanjing, Pinghe, and Yunxiao were above 0, indicating slight As pollution in these districts. The mean *I_geo_* values of the other trace metals were not as high as As, with only Yunxiao and Licheng districts polluted by Pb and Zn, respectively, and the mean *I_geo_* values of Cr and Ni were negative in all districts. The calculations for *l_geo_* indicate that the majority of soil samples presented a low ecological risk from trace metals, suggesting that the ecological risk from trace metals in Golden Triangle soils is generally low.

RI considers both metal enrichment and toxicity and was chosen as an integrated score to evaluate all trace metals in this study. To identify overall pollution levels and locations with high ecological risk, it is necessary to map the RI using IDW interpolation (Figure 4). The spatial distribution of RI shows that most of the study area did not exceed a level of 150, suggesting that the potential ecological risk from these six elements in the Golden Triangle of Southern Fujian Province is low. Spatial mapping of RI also revealed that Xiamen had the lowest average RI value (25.7), followed by Quanzhou (29.3) and Zhangzhou (34.2). Moreover, the spatial pattern revealed that Zhao’an had the most severe RI hot spot. In areas with an RI value greater than 100, the soil concentrations of As contributed more than 60% to the value, suggesting that As pollution is more severe than that from other metals. The RI values of soils in 28 districts of the Golden Triangle also showed that the average RI values of these districts were all below 50 (Appendix A), and As contributed most to the whole RI value in every district, indicating that As pollution was an important aspect to be considered in the Golden Triangle region.

### 3.3. Assessment of Human Health Risk from Trace Metals in Golden Triangle Soils

The noncarcinogenic risk assessment of six trace metals in Golden Triangle soils due to three different contact pathways are shown in Table 2. The results imply that the risk of adult exposure to Cr, Cu, Ni, Pb, and As in urban soils is low and within safety standards, with all HQ mean values below one. However, for children, several HQ values for As were above 1, indicating a cancer health risk from As exposure in the study region. The mean HQ values for adults exposed to As, Cr, Cu, Ni, Pb, and Zn were 4.69 × 10^−2^, 1.33 × 10^−2^, 6.87 × 10^−4^, 7.30 × 10^−4^, 1.65 × 10^−2^, and 3.86 × 10^−4^, respectively, and for children were, 3.06 × 10^−1^, 8.87 × 10^−2^, 4.79 × 10^−3^, 5.09 × 10^−3^, 1.15 × 10^−1^, and 2.69 × 10^−3^, respectively.

Because not all trace metals have SF values, we calculated the carcinogenic risks (CR) of As, Cr, and Ni in the Golden Triangle soils (Table 2). The mean CR values for As, Cr, and Ni for adults were 1.21 × 10^−7^, 9.65 × 10^−7^, and 8.89 × 10^−9^, respectively, and for children were, 8.48 × 10^−7^, 6.75 × 10^−6^, and 6.22 × 10^−8^, respectively. The CR values for As, Cr, and Ni were consistently less than 1.0 × 10^−6^, indicating that the CR is within safety levels and therefore negligible. Zhangpu had the highest mean HQ values for As and Pb, and the highest CR mean values for As for both adults and children, while the highest HQ and CR mean values for Cr, Cu, Ni, and Zn were found in Jimei, Yongchun, Zhangpu, and Licheng.

The results of the assessments for HI through three contact pathways for both adults and children and for TCR showed that the average HI and TCR values were below 1 and 10^−5^, respectively, indicating a mild threat of human health effects from the six trace metals in the Golden Triangle soil samples (Figure 5). The HI values for soil trace metals in the three cities are as follows: Zhangzhou > Quanzhou > and Xiamen, and CR values are as follows: Xiamen > Zhangzhou > and Quanzhou. The highest mean HI value of metals for adults and children were in Yunxiao soils (0.1165 for adults and 0.7739 for children) and the lowest were in Xiang’an (0.0516 for adults and 0.3444 for children). The highest TCR value of metals for adults and children were in Haicang soils (1.82 × 10^−6^ for adults and 1.27 × 10^−5^ for children) and the lowest were in Dehua (3.95 × 10^−7^ for adults and 2.77 × 10^−6^ for children). Compared with children, both HI and TCR values for adults obtained in our study were all below threshold values, reflecting that trace metals impact adults’ health slightly. However, there were 17 soil samples that have the potential to pose noncarcinogenic risks to children (HI > 1); all of these soil samples were collected from Zhangzhou City, indicating that more attention should be paid to this area. It is noteworthy that the TCR values of 23.3% of soil samples for children exceeded 1.0 × 10^−5^, which should be considered a carcinogenic risk to children. In these excess samples, 61.3% of them were collected from Xiamen City, because of the high concentration of Cr. The CR value of Cr contributed nearly 80% to the TCR value, reflecting Cr may pose a higher lifetime carcinogenic risk to children via ingestion pathway compared with As or Ni. The TCR values of four districts for children in Xiamen (Haicang, Jimei, Huli, and Tong’an) were significantly higher than most districts in Zhangzhou and Quanzhou. Thus, it is necessary to take measures to control the acceleration of trace metals, especially Cr in Xiamen.

### 3.4. Source of Trace Metal Contamination using PCA and Matrix Cluster Analysis

The results show a high correlation between Cr, Cu, Ni, and Zn, indicating that the four elements were obviously derived from the similar human activities, such as industrial activities (Table 3). In contrast, the correlation between Pb and Cr, as well as Ni, was weak, suggesting that Pb might originate from another source. Conversely, Pb significantly correlated with As, due to their similar origins. Cu and Zn significantly correlated with other four elements, indicating that Cu and Zn might be determined by multifold aspects.

Generally speaking, the results of the PCA and matrix cluster analyses are consistent and can be discussed together. PCA was used to analyze the concentrations of the six metals and the results are presented in Table 3. The first and second principal components with eigenvalue was greater than 1 (PC1 and PC2) were extracted and the accumulated variance explained 67.07% of the total variation [13]. The contribution rate of first principal component (PC1) was greater than 43% of the total variation and mainly consisted of four metals (Cr, Cu, Ni, and Zn), with variance values 0.645, 0.832, 0.793, and 0.753, respectively, indicating that the main sources of these four metals are associated with each other. The PC2 accounted for 23.99% of cumulative variance and had positive loadings (0.722 and 0.448) of Pb and As, implying that Pb and As had a common source. Matrix cluster analysis was used to generate a heatmap of the concentrations of the six metals (As, Cr, Cu, Ni, Pb, and Zn) in the 456 soil samples and to classify the metals and soil samples into different groups (Figure 6). The concentration of each metal in a single soil sample was represented in each rectangle of the heatmap. In the heatmap, red colors represent high metal concentrations and blue colors represent low concentrations. For red colors, as the grid color lightens, the metal concentration decreases, oppositely, for blue colors, as the grid color lightens, the metal concentration increases. Along the x-axis (Figure 6), the six elements are divided into two groups. Cu, Ni, Cr, and Zn are clustered in the same group, As and Pb were clustered together in a different group, consistent with the results of PCA.

On the y-axis, the heatmap organized the 456 soil samples into five groups: group I, II, III, IV, and V. Appendix A lists the average concentrations of the six metals in each of the five groups. To improve our analysis, we summarized the land use types of the five groups based on the heatmap classification (Appendix A). Group III includes only 4 samples; the colors for Cr and Ni are crimson and suggest that the average concentrations of Cr and Ni in group III were higher than the other groups (*p* < 0.05). The sites of these samples were on the edge of Zhangzhou City and close to each other, demonstrating that Cr and Ni pollution was serious in this area. Two samples of group III were collected from the machinery manufacturing industrial area, these factories might be the pollution source. Group V is the largest group with 200 samples, while group I, II and IV contain 44, 180, and 28 soil samples, respectively. For most metals, especially Cr, Cu, Ni, Pb, and Zn, the sources are complex and diverse and are normally associated not only with organic matter but also with human activities such as diesel fuel and industrial installations [2,9,39]. The concentrations of Cr, Cu, and Ni in the group V samples were markedly higher than those in the group I, II, and IV samples (*p* < 0.05). In addition, group V had high concentrations of Pb and Zn as well. Of these groups except group III, group V had the highest proportion of industrial and roadside area and the lowest proportion of forested and agricultural land, indicating that Cr, Cu, Ni, Pb, and Zn are vulnerable to the effects of anthropogenic activities and are associated with industrial and traffic activities, which is consistent with previous studies [4,10,40]. The results of Appendix A show that Haicang, Jimei, and Huli districts had the highest Cr and Ni concentrations, these three districts are all in Xiamen and famous for their electronics industry, which may be a common source of Cr and Ni in the soils [41,42,43]. Group IV had the highest As and Pb concentration, and nearly 40% of these samples were collected from forest land in the southwest of the Golden Triangle. Pb is usually regarded as an anthropogenic metal and has been found to accumulate in urban centers due to high levels of automotive exhaust emissions [44,45]. However, in our study, Pb was also found to accumulate in the undeveloped areas in Zhangzhou with low road density and high elevation, suggesting that the accumulation of Pb was affected by another source. Steinnes et al. had inferred that atmospheric transport strongly influenced the accumulation of pollutants such as Pb, As, Sb, and Cd in forest surface soils [46]. Brännvall et al. also found that air pollution and not natural processes was the main source of Pb in the surface soils of forests in Sweden [47]. Thus, we considered that atmospheric deposition was the dominant factor evaluated the contents of As and Pb in the soils of some districts [48,49]. It has been reported that Zn is commonly used in fertilizers and pesticides [50]. Besides, sewage irrigation could elevate the contents of Zn in the agricultural soils as well [51]. Therefore, the high average contents of Zn in group I could ascribe to agricultural activities. Compared with other groups, the contents of six elements in group II were generally lower than those in other groups. These samples were collected mainly from agricultural, residential and public areas, representing agricultural and residential pollution. Thus, various anthropogenic activities caused the high concentration of six metals in many districts. The developed districts such as Siming, Huli, Haicang, Jimei, Fengze, Licheng, Jinjiang, Shishi, Longwen, and Dongshan will be easily affected by industrial, traffic, and residential activities. The soil of undeveloped districts with vast tracts of farmland and forest such as Tong’an, Xiang’an, Luojiang, Quangang, Hui’an, Anxi, Yongchun, Dehua, Nan’an, Xiangcheng, Yunxiao, Zhangpu, Zhao’an, Changtai, Nanjing, Pinghe, Hua’an and Longhai may be mainly polluted by atmospheric deposition, residential, and agricultural activities.

## 4. Conclusions

We carried out an investigation to determine the concentrations and spatial patterns of six trace metals (As, Cr, Cu, Ni, Pb, and Zn) in topsoil samples from the Golden Triangle of Southern Fujian Province. Average concentrations were ranked from highest to lowest: Zn, Pb, Cr, Cu, Ni, and As. In relation to the reference background values in Fujian Province soils, the mean concentrations of As, Pb, and Zn in the Golden Triangle soils were elevated, while Cr, Cu, and Ni were lower than the reference samples. The spatial characteristics of these metals confirmed that the topsoil of developed areas accumulated Cr, Cu, Ni, and Zn, whereas As and Pb accumulated in undeveloped areas.

The *I_geo_* analysis of individual trace metals revealed that Cr had the lowest value and therefore Cr pollution was slight, whereas As had the highest value, indicating severe pollution. Based on the RI results, most of the study area did not exceed a level of 150, indicating that the potential ecological risk from trace metals in the Golden Triangle is within acceptable levels. The health risk assessment model was employed to determine human health risks from three pathways of trace metals exposure from soils in the Golden Triangle. Both HI and TCR values for adults were all at the safe levels, reflecting that trace metals pose slightly non-carcinogenic and carcinogenic risk to adults’ health. However, the TCR values of more than 20% soil samples were higher than the threshold values (1.0 × 10^−5^), suggesting that children are facing carcinogenic health risk of metals especially Cr via ingestion pathway. In addition, the carcinogenic risk for children in Xiamen was significantly higher than most districts in Zhangzhou and Quanzhou.

The trace metal source identification method included PCA and matrix cluster analysis. This method showed that the enrichment of Cr, Cu, Ni, and Zn was associated with similar anthropogenic sources, while Pb and As pollution was mainly influenced by atmospheric deposition. Agricultural activities played an important role in accumulation of Zn as well. Meanwhile, traffic activities might be another potential source of Pb accumulation.

These findings reveal that trace metal pollution in Golden Triangle soils is slight. However, continued monitoring and informed policy decisions should be carried out to improve the soil environment. In addition, as children are more sensitive and vulnerable to the health risks posed by trace metals, especially As and Cr, more effective measures should be taken to prevent children’s exposure to trace metals.

## Figures and Tables

**Figure 1 ijerph-16-00097-f001:**
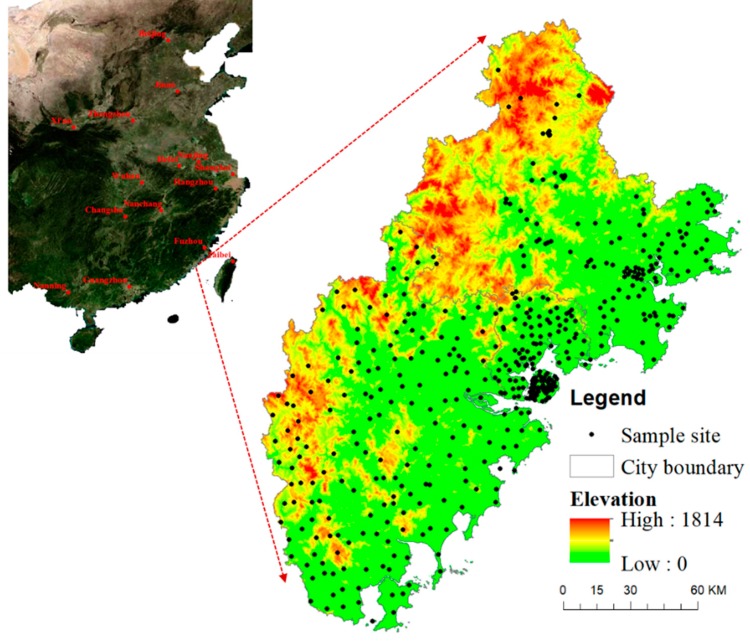
Distribution of sampling sites in the Golden Triangle of Southern Fujian Province.

**Figure 2 ijerph-16-00097-f002:**
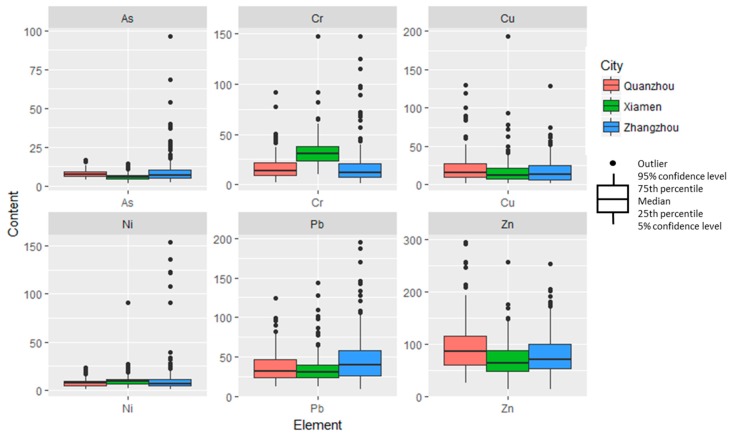
Heavy metal concentrations in the soils of Quanzhou, Xiamen, and Zhangzhou.

**Figure 3 ijerph-16-00097-f003:**
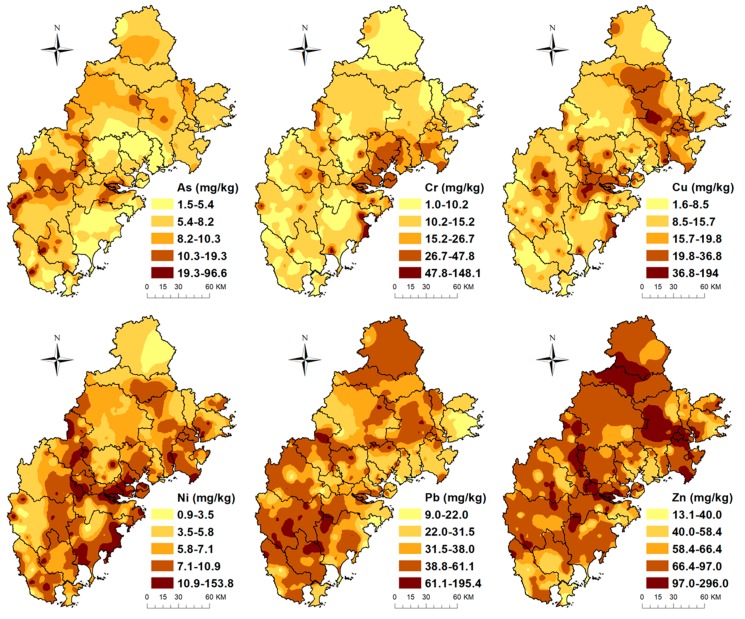
Spatial distribution maps of the concentrations of six trace metals in the soils of the Golden Triangle.

**Figure 4 ijerph-16-00097-f004:**
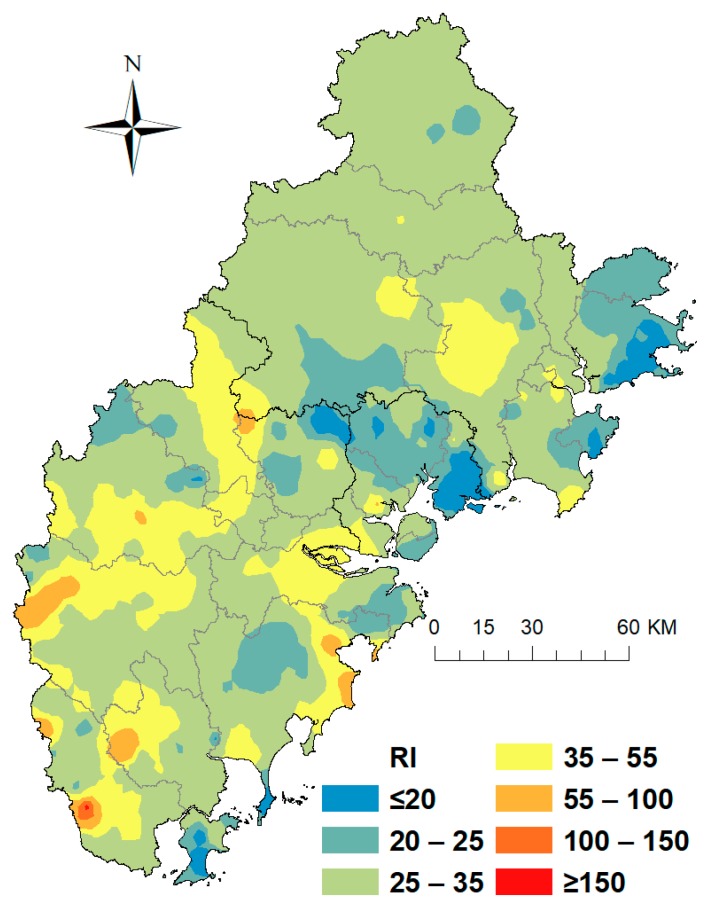
Spatial distribution maps of RI in the soils of the Golden Triangle.

**Figure 5 ijerph-16-00097-f005:**
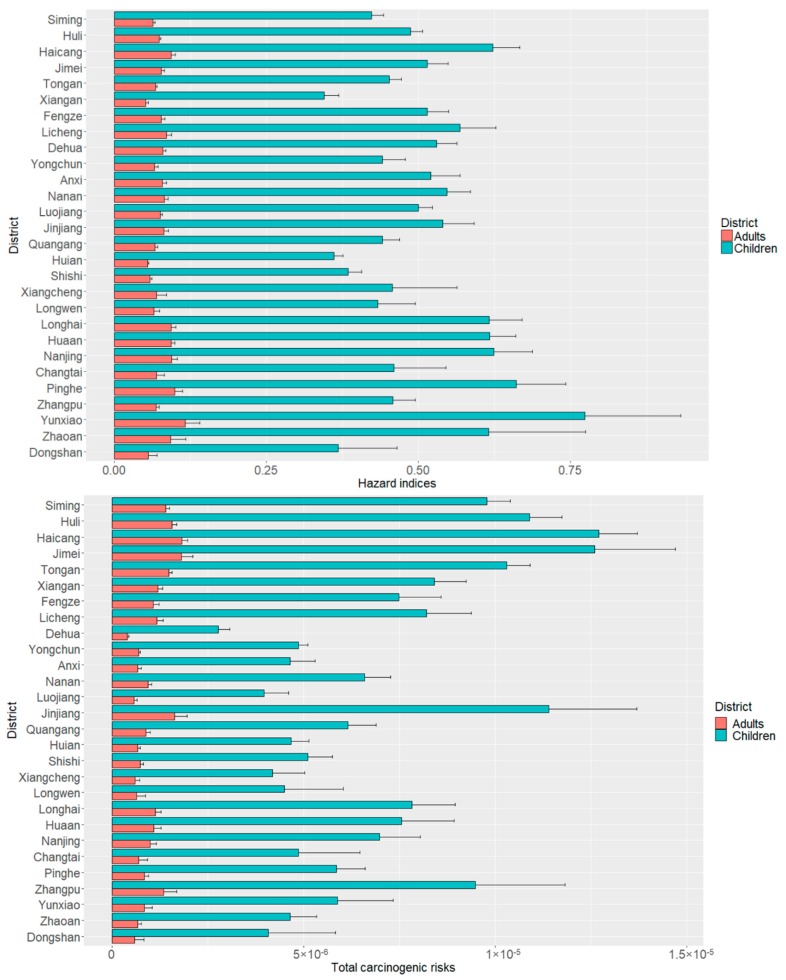
HI and TCR values for trace metals in the soils of different Golden Triangle districts.

**Figure 6 ijerph-16-00097-f006:**
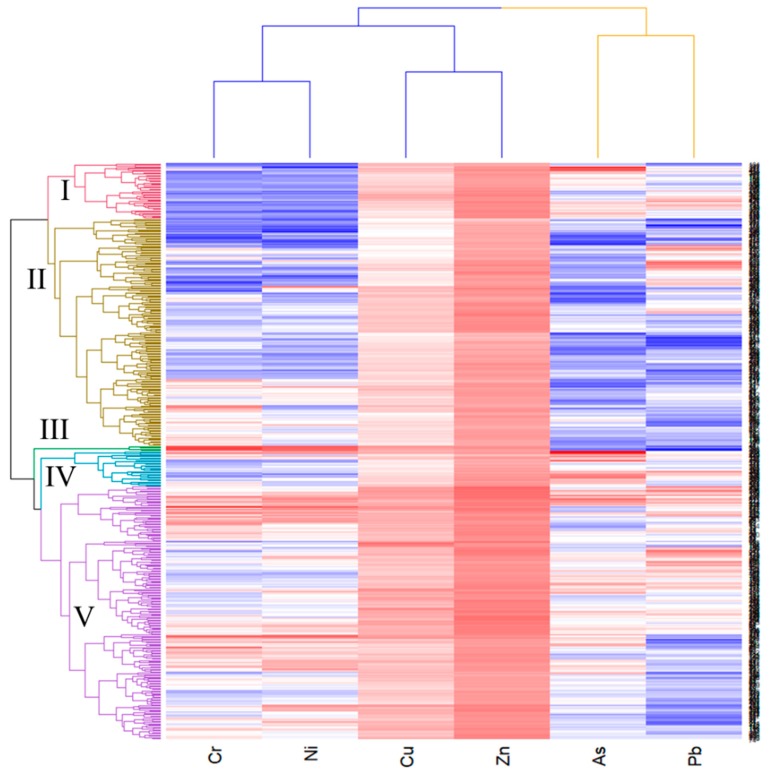
Clustering tree of matrix for cluster analysis of trace metals in the soils of the Golden Triangle.

**Table 1 ijerph-16-00097-t001:** Descriptive statistics of trace metals in the soils of the Golden Triangle (mg·kg^−1^).

	As	Cr	Cu	Ni	Pb	Zn
Mean	7.97	22.80	19.79	10.51	41.14	82.92
Median	6.40	18.30	13.60	7.90	33.35	71.95
Standard deviation	7.15	19.40	20.04	14.69	26.17	44.55
Minimum	1.50	1.00	1.60	0.90	9.00	13.10
Maximum	96.60	148.10	194.00	153.80	195.40	296.00
Background values of Fujian soil	5.78	41.30	21.60	13.50	34.90	82.70

**Table 2 ijerph-16-00097-t002:** Health risks from trace metals in Golden Triangle soils.

Adults	HQ_ing_		HQ_derm_		HQ_inh_		CR_inh_	
	Mean	SD	Mean	SD	Mean	SD	Mean	SD
As	3.64 × 10^−2^	3.26 × 10^−2^	1.05 × 10^−2^	9.45 × 10^−3^	6.47 × 10^−5^	5.81 × 10^−5^	1.21 × 10^−7^	1.09 × 10^−7^
Cr	1.04 × 10^−2^	8.86 × 10^−3^	2.08 × 10^−3^	1.77 × 10^−3^	8.03 × 10^−4^	6.83 × 10^−4^	9.65 × 10^−7^	8.21 × 10^−7^
Cu	6.78 × 10^−4^	6.86 × 10^−4^	9.01 × 10^−6^	9.13 × 10^−6^	4.96 × 10^−7^	5.02 × 10^−7^		
Ni	7.20 × 10^−4^	1.01 × 10^−3^	1.06 × 10^−5^	1.49 × 10^−5^	5.14 × 10^−7^	7.18 × 10^−7^	8.89 × 10^−9^	1.24 × 10^−8^
Pb	1.61 × 10^−2^	1.02 × 10^−2^	4.28 × 10^−4^	2.72 × 10^−4^	1.18 × 10^−5^	7.49 × 10^−6^		
Zn	3.79 × 10^−4^	2.03 × 10^−4^	7.55 × 10^−6^	4.06 × 10^−6^	2.78 × 10^−7^	1.50 × 10^−7^		
**Children**	**HQ_ing_**		**HQ_derm_**		**HQ_inh_**		**CR_inh_**	
	**Mean**	**SD**	**Mean**	**SD**	**Mean**	**SD**	**Mean**	**SD**
As	2.55 × 10^−1^	2.29 × 10^−1^	5.17 × 10^−2^	4.64 × 10^−2^	4.53 × 10^−4^	4.07 × 10^−4^	8.48 × 10^−7^	7.61 × 10^−7^
Cr	7.29 × 10^−2^	6.20 × 10^−2^	1.02 × 10^−2^	8.68 × 10^−3^	5.62 × 10^−3^	4.78 × 10^−3^	6.75 × 10^−6^	5.74 × 10^−6^
Cu	4.75 × 10^−3^	4.81 × 10^−3^	4.43 × 10^−5^	4.48 × 10^−5^	3.47 × 10^−6^	3.51 × 10^−6^		
Ni	5.04 × 10^−3^	7.04 × 10^−3^	5.22 × 10^−5^	7.30 × 10^−5^	2.10 × 10^−5^	2.94 × 10^−5^	6.22 × 10^−8^	8.70 × 10^−8^
Pb	1.13 × 10^−1^	7.17 × 10^−2^	2.10 × 10^−3^	1.34 × 10^−3^	1.41 × 10^−5^	8.96 × 10^−6^		
Zn	2.65 × 10^−3^	1.42 × 10^−3^	3.71 × 10^−5^	1.99 × 10^−5^	1.95 × 10^−6^	1.05 × 10^−6^		

**Table 3 ijerph-16-00097-t003:** PCA loadings and correlation coefficients of trace metals in the soils of the Golden Triangle.

Element	Principal Component Analysis		Correlation Analysis					
	PC1	PC2	As	Cr	Cu	Ni	Pb	Zn
As	0.477	0.448	1	0.074	0.285 **	0.207 **	0.295 **	0.313 **
Cr	0.645	−0.609	0.074	1	0.405**	0.726 **	−0.059	0.182 **
Cu	0.832	−0.007	0.285 **	0.405 **	1	0.547 **	0.152 **	0.650 **
Ni	0.793	−0.442	0.207 **	0.726 **	0.547 **	1	0.03	0.365 **
Pb	0.344	0.722	0.295 **	−0.059	0.152 **	0.03	1	0.419 **
Zn	0.753	0.382	0.313 **	0.182 **	0.650 **	0.365 **	0.419 **	1
Eigenvalue	2.59	1.44						
% of Variance	43.09	23.99						
Cumulative %	43.09	67.07						

* At the 0.05 significance level. ** At the 0.01 significance level.

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
