# Peer review of "Distribution and Health Risk Assessment of Trace Metals in Soils in the Golden Triangle of Southern Fujian Province, China"

_ijerph, 2018, doi:10.3390/ijerph16010097_

Round 1

Reviewer 1 Report

General comments

This manuscript explores and determines many levels and index regarding heavy metal contamination and ecological and health risk assessment. The analysis made by the authors is extensive and thorough.

However, although this manuscript is good as a whole, there are several aspects that the authors need to check.  

Firstly, it is important for the authors to justify their sampling. The area they pretend to analyse is very big (25315.39 km2) so I am a bit concerned about the sampling followed and the amount of soil samples they included. I would like to see how they justify and explain this, since I believe that in order to make an accurate study of the area and the elements and aspects they want to study the amount of soil samples they should take must be much higher.

And secondly, the discussion of the results is a bit unclear. Authors should improve the structure of this and its content, trying to relate the results of the main heavy metals identified in each area or district with the main soil uses present in this area. They make an initial attempt in Table S4. However, this has to be extended to much of the discussion made. Another important aspect to consider is I would prefer if results and discussion were united, making it easier to see the patterns observed by the authors in the results and the interpretation of them.

Abstract

Please check some of the English in this section.

            - Line 13: “intensified” instead of “intensifying”

            - Line 16: “of” not “from”

Also, indicate briefly what the authors think the 5 groups identified by the cluster analysis stand for, what represents each of them.

Introduction

- Line 43-44: Is this a general pattern seen worldwide or does it only belong to China? The references included only refer to China.

            - Line 48: indicate entering pathways please

- Line 63: delete “others” and specify which are considered common by the authors.

- Line 69: check English – “depend on” instead of “depends”

- Line 84-85: Please explain in the previous sentences why you selected this group of Heavy Metals (HM) and why each of them is important. Also indicate the ones that are initially considered to be more toxic (Eg. Pb)

- Line 89: justify the final selection of these indexes.

Material and Methods

- Lines 108-114: There are some issues that need to be clarified regarding the sampling. Firstly, the depth of the subsamples taken from the 16 m2 needs to be indicated. Secondly, it is important to make clear how the authors selected the sampling points, as the area covered is very big. And thirdly, please justify the nº of samples and how are these enough to cover the study area and to achieve the goals presented.

- Lines 115-121: This section regarding the digestion and analysis of soil samples needs more detail. Authors should add reference to justify the methods selected. They should also indicate if the wet digestion was in heatplate or microwave assisted. Finally they should also add reference to justify the adequacy of the precision in the measures and indicate the certified standard soils used.

- Lines 143-144: I reckon 7 classes is too much, making interpretation difficult. Please reduce. Also, Table 1 does not show the classes identified; its table S1 the one that shows this. Please check.

- Line 152: Which classes? Table S1 again? 

- Lines 166-167: please add reference to justify these reference values.

Results 

As general comment previous to all the detailed ones, I reckon it is important to point out two aspects: firstly, that this section would be much better if its was united with the discussion section. This section contains a lot of results which a believe would be easier to interpret and follow if the discussion was included after them. And secondly, it is difficult to follow when the authors speak about districts and cities, and what districts belong to what city, etc. Please revise and keep the same criteria throughout the whole manuscript. 

- Line 193: Please indicate if these are high or low according to the authors and justify the answer.

- Line 196: Please indicate where do these reference values come from.

- Lines 199-201: Can these differences be justified? Maybe main soil uses/activities in the area)

- Fig 3: Check the legend. Too many classes and unevenly distributed.

- Lines 295- 297: delete and introduce in the Introduction section

- Lines 303-311: please indicate how did the authors select this amount of PCs, following which criteria.

Discussion

This section is good as a whole although it would be interesting to: 

- Put together with results

- Further explain and summarise how the origin of HM and area meet with the values determined in each district

- Further explain the carcinogenic risk for children

Conclusions

            Please include firther conclusions regarding the carcinogenic risks for children (areas, values, etc.)

I hope these suggestions contribute to improve the quality of the manuscript.

Reviewer 2 Report

Dear authors,

your study contains a very large dataset. I like the manuscript, however, I miss the basic properties of the studied soil, such as organic matter content, pH, soil type. These parameters are necessary to predict mobilization and potential toxicity of the trace metals. Please provide and discuss this if possible.  I recommend including more international papers.

Many regards

Reviewer

Reviewer 3 Report

@page { margin: 2cm } p { margin-bottom: 0.25cm; line-height: 120% }

As is not a metal

The aim(s) should be expressed in the form of a testable hypothesis

L. 108 - What about sampling design? it seems preferential sampling, which hampers any statistical inference

L. 111 – mixing the four subsamples into one single composite sample mens that pseudo-replication has been achieved, that is a statistical flow

L. 112 – soils sealed in plastic bags: hopefully just for transportation to the lab

L. 119 - "The analyzed precision was between 5% and 6% and did not exceed 8%" this is an odd sentence: I guess precision was 5-8%

L. 121 – what about element recoveries?

I’m confident colors in fig 1 refer to elevation, please be clearer

L. 125 – what about data frequency distribution? (from tab 1 it seems that they are not normally distributed, so all statistical analysis is not valid) please also consider the issue(s) above about statistical flows; how many replicated measurements were taken for a site?

L. 130 – I really doubt Anova can be used for any co rrelative study

L. 136 – reasons for using IDW instead of kriging?

L. 142 - what background values were used for calculating Igeo? these are fundamental

the link www.mdpi.com/xxx/s1 does not work, so I could not check supplementary material

Fig. 2 confirms the non-normal data distribution

All results are probably based on a wrong statistical approach, so also the conclusions are not warranted.

Reviewer 4 Report

The authors provide measurements of heavy metal concentrations from soil samples in the Golden Triangle region and estimate health risks according to the USEPA model. It is well written and the methods as well as the findings are described clearly. The manuscript lacks of a comparison of these findings with real world health data. I would really appreciate it to see health data from public registry in the region (i.e. hospital admissions subdivided for reason) in order to see, whether the calculated risks withstand a comparison to real world data.

Round 2

Reviewer 3 Report

@page { margin: 2cm } p { margin-bottom: 0.25cm; line-height: 120% }

Although the paper is now much improved, the Authors replayed to my issues only partially; below a list of points still to be solved

point 2 - the aim is not yet expressed in the form of a testable hypothesis; the AA stated that they “didn’t propose a hypothesis before analysis” and this is really odd: how can a scientist plan an experiment without any hypothesis to be tested? the fact that the AA answered that they wish only to “explore the ecological and human health risks and to identify the potential sources” prompt for a descriptive paper

point 3 – the sampling design, and the reasons for such, have to be included into the methods

point 4 – the AA mad a mistake: to avoid pseudoreplication the subsamples should have been analysed separately

point 12 – the choice of background values is key in this and similar studies since Igeo and RI are based on such values; the AA failed to explain how these values were identified as background; the fact that most Igeo values were below zero further confirm this flaw

Author Response

Dear reviewer,

We have answered the questions and thank you for these suggestions to improve our manuscript quality and help us to improve the experimental design in the further study! 

Many regards

Authors

Reviewer 4 Report

Dear authors,

with your statement in mind, I look forward to your future study including public health data:

> We will continuously monitor soil heavy metal pollution and record the data to test correlations
> with public health data in the further study.

For now I vote for 'accept' because the data from your manuscript is important for future analyses.

Good luck and regards...

Author Response

Dear reviewer,

Thank you for your suggestion! And we hope we can make a comprehensive research about the relationship between soil heavy metals and public health in the future study.

Many regards

Authors

Round 3

Reviewer 3 Report

In my opinion the AA now replyed in a very honest way, and I'm quite satisfied of their changes; my concern is that now the paper is horribly long, but this is more an editorial issue, so I leave this point open for the final choice of the editor